# The Toxic Masculinity Scale: Development and Initial Validation

**DOI:** 10.3390/bs14111096

**Published:** 2024-11-14

**Authors:** Steven Michael Sanders, Claudia Garcia-Aguilera, Nicholas C. Borgogna, John Richmond T. Sy, Gianna Comoglio, Olivia A. M. Schultz, Jacqueline Goldman

**Affiliations:** 1School of Psychological Science, Oregon State University, Corvallis, OR 97331, USAjacqueline.goldman@oregonstate.edu (J.G.); 2Department of Psychology, University of Alabama at Birmingham, Birmingham, AL 35294, USA; borgogna@uab.edu

**Keywords:** measurement, masculinity, instrument development

## Abstract

The present study sought to develop and validate the Toxic Masculinity Scale (TMS). Following scale development best practices, a thorough review of the literature and existing measures was conducted. Next, a qualitative inquiry using a grounded theory approach was employed to develop a data-driven definition of toxic masculinity and 165 proposed instrument items. These items were reviewed and modified with input from content experts (*N* = 6). The initial 108 items were administered to a preliminary sample (*N* = 683) of U.S. White undergraduate men. Exploratory factor analysis indicated a five-factor structure (i.e., Masculine Superiority, Gender Rigidity, Emotional Restriction, Repressed Suffering, Domination and Desire). Item analysis yielded a 35-item five factor survey that was administered to a second novel sample (*N* = 408) of White undergraduate men. A confirmatory factor analysis indicated inadequate fit for the 35-item scale; however, fit was improved by reducing scale length to 28 items loading onto 4 factors (removal of the Domination and Desire factor). Internal consistency reliability, construct validity, and discriminant validity were explored with this sample. The TMS28 demonstrated strong positive correlations with related measures (e.g., CMNI, MRNI) and a strong negative correlation with a diametrically opposed measure (i.e., LFAIS). Additionally, the TMS28 demonstrated strong adequate internal consistency for the scale overall (*α* = 0.93) and for the four subscales (i.e., *α* = 0.87–0.94). Future directions and implications for the instrument are discussed.

## 1. Public Significance Statement

Toxic masculinity is a visible construct in the psychology of men. However, proxy instruments have been used historically for measurement. The present study provides a theory-driven definition of toxic masculinity and a validated instrument to measure the construct.

## 2. The Toxic Masculinity Scale: Development and Initial Validation

Toxic masculinity as a construct was coined in the 1980s by Shepherd Bliss to characterize his father’s authoritarian masculine behaviors from his days in the military (no specific behaviors were listed; [1]. Although the term has existed since the late 20th century, it has only relatively recently become popular in academic texts. Since approximately 2013, feminist theorists and scholars have attributed undesirable behaviors by men such as misogyny, homophobia, and physical violence to toxic masculinity [2]. For masculine traits and behaviors to be perceived as *toxic*, they must be damaging to the person and/or the people around them. Otherwise, those traits and behaviors are only characteristics of endorsing gender roles shaping the presentation of masculinity [3]. Traits such as seeking domination and power, being work-focused, taking risks, and toughness are examples of traits/behaviors associated with masculinity in typical instruments (e.g., [4,5,6]). The Conformity to Masculine Norms Inventory (CMNI) is a series that measures the conformity to those masculine norms that emphasize gender roles. In addition, this instrument is grounded in the work of Mahalik [5] on those behaviors that emphasized in the socially dominant groups that shape the gender norms that are imbedded into our society [6]. Moreover, the Male Role Norms Inventory (MRNI) zooms in on traditional masculinity ideologies and beliefs about men. This instrument aims to capture attitudes towards traditional masculinities in general, not of oneself. In addition to the aforementioned traits, other researchers often include heterosexism and misogyny as facets of toxic masculinity in their studies [7]. The term ‘toxic masculinity’ has been employed to critique a strict attachment to societal-based masculinity traits in hopes of reversing these gender norms to encourage more pro-social behavior [3]. The American Psychological Association (APA) released a set of guidelines for those who work with men and boys, and more specifically how to work with those who present the notion of “traditional masculinity”. They attributed characteristics of masculinity such as dominance and aggression to gender attitudes that manifest more often in men [3].

Toxic masculinity typically gets referenced in tandem with feminist and womanist ideals and constructs. The term has been adopted by feminists as an easier way to discuss homophobia and misogyny in male behavior [1]. Toxic masculinity has become a way to quickly characterize a person based on a few different attitudes they express [1]. It is suggested that the idea of “traditional masculinity” is both others and those who subscribe themselves to those gender constructs [3].

The definition of toxic masculinity fluctuates depending on context. For example, hegemonic masculinity, sometimes used as proxy for toxic masculinity, is a manifestation of masculinities that is characterized by the enforcement of restrictions in behavior based on gender roles that serve to reinforce existing power structures that favor the dominance of men (e.g., [7,8,9]). Hegemonic masculinity speaks to the systems and processes that elevated men to positions of power and maintain their dominance (e.g., [10,11]). Additionally, traditional masculinity is marked by stoicism, competitiveness, dominance, and aggression, characterizing it by an adherence to gendered attitudes [3]. In multiple studies examining the construct of toxic masculinity, instruments designed to assess the adherence to, and/or endorsement of, masculine traits are used as proxy toxic masculinity measures. In social science research, validity is concerned with whether instruments measure what they intend to measure [12]. It is important to study toxic masculinity to establish its prevalence as well as explore how the presence of toxic masculinity affects men’s psychological well-being. In addition, the study of toxic masculinity may provide an insight on how the presence of these traits affect men’s relationships not only with intimate partners but with family and themselves. In order to accurately and effectively study toxic masculinity, the instruments used by researchers need to be valid and reliable.

## 3. The Current Research

The overarching aim of the present series of studies was to develop and initially validate an instrument to assess the presence of toxic masculinity in White men in college. Focusing on White men to validate this instrument was crucial as traditional masculinity is often attributed to White masculine elites [3]. The primary research objectives were threefold: (1) Using qualitative open-ended questions, develop a data-driven definition of toxic masculinity and potential items for the toxic masculinity instrument; (2) use a cross-sectional sample of White men currently enrolled in college to assess the strength of the proposed items and factor structure; and (3) use a *different* sample of White men enrolled in college to assess the psychometric properties of the proposed toxic masculinity scale, including the revised factor structure, internal consistency, test-retest reliability, and construct validity. The theoretical framework for the instrument development process was informed by first-phase qualitative approaches. We intentionally followed the data to avoid bias from the research team and to allow a natural evolution of the instrument items, essentially using a grounded theory approach. A grounded theory approach seeks to explain a process experienced by many people that no prior theory has satisfactorily explained [13,14].

### 3.1. Study 1: Scale Item Development

The purpose of Study 1 was to develop a working definition of *toxic masculinity* and the Toxic Masculinity Scale (TMS) pool items. This was achieved through qualitative inquiry (i.e., thematic coding) and review from content experts to establish face validity of the proposed items.

#### 3.1.1. Participant Characteristics

Participants were 721 undergraduate students from a large public university from the Pacific Northwest who identified as White. Participants were recruited through the school’s SONA systems and consisted of 197 males, 494 females, 23 non-binary students, and 6 gender fluid students. Sona is an online participant recruitment and management system used by more than 900 colleges and universities worldwide [15]. Participants’ ages ranged from 18 to 54 (*M_age_* = 21.32, *SD*_age_ = 5.76). There were an additional 88 White undergraduate participants recruited through the online crowdsourcing platform Prolific. Prolific is a different online research platform that matches researchers with eligible participants as an alternative to the Amazon Mechanical Turk (MTurk) research platform [16]. Peer and colleagues determined that while Prolific’s response rate was slightly lower than MTurk, it was faster than a university sample (e.g., recruited via Sona). Additionally, their research found that Prolific samples replicated existing results and delivered higher quality data than either a university sample or an MTurk sample. The Prolific version of the sample included 41 males, 42 females, and 5 nonbinary participants. Their ages ranged from 18 to 69 (*M*_age_ = 35.83, *SD*_age_ = 12.24). All participants were provided informed consent and the Oregon State University institutional review board approved the study.

#### 3.1.2. Measure

All participants were administered an online qualitative survey with six open-ended items. These items included “What is your definition of masculinity”, “How would you define “toxic masculinity”, “How can women display traits of toxic masculinity”, “In media, describe instances of toxic masculinity you have seen (if they exist)”, “Give examples of toxic masculinity you have observed in person (if any)”, and “How have you displayed traits of toxic masculinity (if ever)”.

#### 3.1.3. Procedures

After receiving approval from the institution’s IRB, participants were recruited through the institution’s SONA platform and Prolific. All participants were administered the survey instrument using the Qualtrics platform. Participants recruited through SONA were awarded class credit as compensation while those recruited through Prolific were immediately compensated USD 4.25 once the survey was completed.

#### 3.1.4. Data Analysis

The qualitative data were initially analyzed by the research team and a working definition of *toxic masculinity* and potential instrument items were generated via discussion of prominent themes in the data. Using themes generated from the qualitative data, the research team generated the following definition of toxic masculinity, which guided the succeeding studies: “*toxic masculinity is the overemphasis and indulgence in societal-based masculine traits to an extent that is harmful to oneself and/or others*”. A panel of independent experts analyzed the proposed items, and, when appropriate, recommended removal of items. If multiple experts recommended the removal of an item, the item was removed from the item pool.

#### 3.1.5. Results and Discussion

A total of 165 potential items were generated from the initial qualitative analysis. This item pool was sent to six independent experts, along with the definition of toxic masculinity serving as the theoretical framework for the project. These experts were asked to identify items that did not reflect that definition, were inappropriately worded, were loaded, were double-barreled, or were reductive. In total, the experts identified 57 items for removal from the item pool, leaving a total of 108 items (see Appendix A). The 108 items utilized for Phase 2 were deemed acceptable by the ad-hoc panel of subject matter experts and informed the series of research studies focused on factor structure and preliminary item analysis.

### 3.2. Phase 2: Initial Exploratory Factor Analysis

The purpose of Phase 2 was to determine the dimensionality of the scale items and reduce the number of items for parsimony.

#### 3.2.1. Participant Characteristics

This sample comprised of 683 novel White men (*M*_age_ = 28.15, *SD* = 10.17) who were currently enrolled as an undergraduate student and lived in the United States. All participants were provided informed consent, recruited through Prolific, and the Oregon State University institutional review board approved the study. More than a third of the sample (74%) had an income of less than $39,000 (*n =* 506). A total of 129 participants, or 16 percent of the original total, were removed from the sample. These participants were removed for not giving consent (*n =* 6), not answering the gender identity question (*n =* 1), failing the first validation item (i.e., “choose ‘strongly agree’ for this item”; *n =* 91), failing the second validation item (i.e., “choose ‘strongly disagree’ for this item”; *n =* 7), and for not disclosing their age (*n =* 24).

#### 3.2.2. Measure

All participants were administered an online survey that asked for their demographic information (e.g., age, gender identity, race) and provided the 108-item Toxic Masculinity Scale (TMS). Participants were asked to rate their level of agreement with a 5-point response format from 1 (Strongly disagree) to 5 (Strongly agree). Sample items include “If I cry, I am weak”, “a man with a lot of sexual partners is more masculine”, and “men cheating on their partner(s) is natural”. The full scale demonstrated excellent internal consistency in this sample (α = 0.98).

#### 3.2.3. Procedures

After receiving approval from the institution’s IRB, participants were recruited through Prolific. All participants were administered the survey instrument using the Qualtrics platform. Participants recruited through Prolific were immediately compensated USD 3.75 (i.e., USD 11.25/h) once the survey was completed.

#### 3.2.4. Data Analysis

Analyses were conducted using STATA (version 17) and SPSS (version 29). A *p*-value less than 0.05 for Barlett’s Test of Sphericity and Kaiser–Meyer–Olkin (KMO) Measure of Sampling Adequacy greater than 0.50 were used to test for psychometric adequacy of the data for factor analysis [17]. Once the data were assessed to be suitable for factor analysis, exploratory factor analysis was conducted. Exploratory factor analysis (EFA) with Promax rotation with Kaiser normalization were used to determine dimensionality and factor structure. Under each factor, only items with pattern coefficients greater than or equal to 0.65 were retained.

#### 3.2.5. Results and Discussion

Barlett’s Test of Sphericity yielded a *χ*2 of 46,483 (*df* = 5778, *p* < 0.001) and KMO Measure of Sampling Adequacy was 0.96, indicating adequacy of the data for factor analysis. An EFA with Promax rotation with Kaiser normalization with pattern coefficients greater than 0.65 identified 6 factors, which we labeled as “masculine superiority”, “domination and desire”, “gender rigidity”, “emotional restriction”, “repressed suffering”, and “competitiveness”. However, we reduced the factors further because the 6th factor only had two items that loaded onto it (i.e., “I am competitive” and “When I compete with friends, I always take it seriously”) and did not fit within the definition of *toxic masculinity* informed by the data generated via the qualitative grounded theory approach in Phase 1. Please refer to Table 1 for the final 35-item factor matrix.

### 3.3. Phase 3: Confirmatory Factory Analysis, Reliability Analysis, and Preliminary Validation Analysis

The purposes of Phase 3 were to assess the proposed factor structure of the 35-item TMS scale, evaluate the construct validity of the scale, and to examine the test-retest reliability of the scale. Phase three was comprised of two separate administrations of the survey instrument.

#### 3.3.1. Participant Characteristics

The sample for the first administration in Phase 3 included 408 White men (*M*_age_ = 29.49, *SD*_age_ = 10.71) who were currently enrolled as an undergraduate student and lived in the continental United States. All participants were provided informed consent, recruited through Prolific, and the Oregon State University institutional review board approved the study. There was a total of six participants, or 0.01 percent of the original total, removed from the sample for providing a duplicate response. In each of these duplicate cases, the participants started the survey instrument, declined to finish after the demographic section, and returned to submit a completed survey. Only the second completed survey was used for the data analysis. In the sample, 336 participants (82%) reported some sort of employment (e.g., full-time, self-employed), 236 participants (58%) reported an income of less than $50,000, and the overwhelming majority of participants identified as straight (*n =* 301), with 32 identifying as gay, 64 identifying as bisexual, 3 as asexual, and 8 as pansexual.

The sample for the second administration included 283 White men (*M*_age_ = 30.12, *SD*_age_ = 10.55) who were currently enrolled as an undergraduate student and lived in the continental United States. All participants were provided informed consent, recruited through Prolific, and the Oregon State University institutional review board approved the study. To be eligible for participation in the second administration, participants had to have completed the first administration. To protect the identity of participants, Prolific extended interviews to all participants in Administration One to return for Administration Two through their platform. All 408 participants from Administration One were eligible to participate in Administration Two. There were no participants removed from the sample. In the second administration’s sample, 235 participants (83%) reported some sort of employment (e.g., full-time, self-employed), 146 participants (52%) reported an income of less than $50,000, and the overwhelming majority of participants identified as straight (*n =* 219), with 25 identifying as gay, 34 identifying as bisexual, 1 as asexual, and 4 as pansexual.

#### 3.3.2. Measures

##### Demographic Information

All participants were administered an online survey that asked for their demographic information (i.e., age, gender identity, race, level of education, income, employment status, and sexual orientation).

##### Toxic Masculinity

The novel Toxic Masculinity Scale (TMS) consisted of the 35 items listed above. Participants were asked to rate their level of agreement with a 5-point response format from 1 (Strongly disagree) to 5 (Strongly agree) on statements such as “If I cry, I am weak”. The scale demonstrated good internal consistency (α’s = 0.93–0.94) and test-retest (*r =* 0.93) reliability between the two administrations.

##### Conformity to Masculine Norms

The CMNI [6] is a 46-item scale that measures adherence to Western masculine norms. This version of the scale is a short form of a 94-item scale by Mahalik and colleagues [5]. The scale had nine subscales, which included emotional control, winning, risk-taking, violence, power over women, playboy behavior, self-reliance, primacy of work, and heterosexual self-presentation. Overall, the scale and subscales demonstrated good validity and reliability (α’s = 0.77–0.91) among men. Participants were asked their level of agreement with a 4-point Likert scale from 0 (Strongly disagree) to 3 (Strongly agree). Scores for the entire scale and each subscale were averaged and higher scores represented greater masculine norm conformity. The scale (α’s = 0.90–0.92) and subscales (α’s = 0.77–0.92) demonstrated good internal consistency within the study sample in both administrations.

##### Male Role Norms

The MRNI [4] is a 21-item scale that measures beliefs and attitudes towards traditional Western masculine ideologies. The scale had seven subscales, which included avoidance of femininity, negativity toward sexual minorities, self-reliance through mechanical skills, toughness, dominance, importance of sex, and restrictive emotionality. Overall, the scale and subscales demonstrated good validity and reliability (α’s = 0.75–0.92) among men. Participants were asked their level of agreement with a 7-point Likert scale from 1 (Strongly disagree) to 7 (Strongly agree). Scores for the entire scale and each subscale were averaged and higher scores represented stronger agreement towards masculine ideologies. The scale (α = 0.96) and subscales (α’s = 0.80–0.93) demonstrated good internal consistency within the study sample between administrations.

##### Liberal Feminist Attitudes

The Liberal Feminist Attitude and Ideology Scale—Short Form (LFAIS; [18]) is a 10-item scale that measures attitudes towards feminist ideologies. The scale demonstrated good validity and reliability (α = 0.81). Participants were asked their level of agreement on a 6-point Likert scale from 1 (Strongly disagree) to 6 (Strongly agree). Scores were summed and higher scores represented stronger agreement towards liberal feminist ideologies. The scale demonstrated good internal consistency (α = 0.87–0.88) within the study sample between administrations.

#### 3.3.3. Procedures

Participants were recruited through Prolific. All participants were administered the survey instruments twice using the Qualtrics platform over a span of six weeks. After completion of the survey, participants were immediately compensated USD 3.75 after completing each administration of the survey for a total compensation of USD 7.50 (i.e., USD 11.25/h). Prior to participant recruitment, the study methods and materials were approved by the Oregon State University Institutional Review Board (IRB), protocol ID HE-2022-15.

#### 3.3.4. Data Analysis

Analyses were conducted using STATA (version 17) and SPSS (version 29). Confirmatory factor analysis was used to test the adequacy of the 5-factor model of the 35-item TMS. An overall χ^2^ with a *p*-value > 0.05, comparative fit index (CFI) > 0.90, Tucker–Lewis Index (TLI) > 0.90, root mean square error of approximation (RMSEA) < 0.05, and standardized root mean square residual (SRMR) < 0.08 were used to test the goodness-of-fit of the model [19]. Convergent validity was assessed by correlating the overall scale and each subscale with the CMNI, MRNI, and LFAIS. Factor loadings must be greater than 0.40 [20].

#### 3.3.5. Data and Study Materials Availability

Data and related study materials are available upon email request to the corresponding author. This study was *not* preregistered.

## 4. Results

Initial CFA for the first administration of the 5-factor model yielded a poor fit. The researchers performed a Pearson correlation with all scale items and found that items 5, 6, 7, 21, and 22 correlated poorly with other items. In addition, items 8 and 9 were dropped as they did not fit the operational definition of toxic masculinity within the study. A second CFA was performed on the 28-item scale that yielded better model fit but did not meet the predetermined criteria for goodness-of-fit. A similar result from the CFA was found in the second administration of the scale. Factor loadings were all greater than the 0.40 threshold in both administrations. Refer to Table 2 for goodness-of-fit indices.

### Convergent and Discriminant Validity

*First administration.* The 28-item TMS scale correlated positively with the CMNI (*r =* 0.73, *p* <.001) and MNRI (*r =* 0.80, *p* < 0.001) and negatively with LFAIS (*r =* −0.81, *p* < 0.001), suggesting good convergent validity. Please refer to Table 3 for correlation matrix of full scales and Cronbach alpha coefficients.

*Second administration.* The 28-item TMS scale correlated positively with the CMNI (*r =* 0.75, *p* < 0.001) and MNRI (*r =* 0.83, *p* < 0.001) and negatively with LFAIS (*r =* −0.82, *p* < 0.001) suggesting good convergent validity. Please refer to Table 4 for correlation matrix of full scales and Cronbach alpha coefficients for the second administration, Table 5 for the correlation matrix between the two administrations, and Appendix B for additional factor loadings.

## 5. Discussion

This multiphase study aimed to define toxic masculinity using a data driven approach, develop an instrument to measure toxic masculinity, and validate that instrument in a sample of White college men. The results of the third phase support the factor structure and validity of the novel instrument, which may provide researchers with the opportunity to measure toxic masculinity with an instrument specifically developed for that purpose.

## 6. Implications

The findings of this study have several research-related implications. Previous research into the construct of toxic masculinity used proxy measures. That is, researchers used standard measures of masculinity and hypothesized that specific scores or endorsements were indicative of thoughts or behaviors representing toxic masculinity. This paper took a novel approach, using best practices to develop an instrument to examine traits, behaviors, and thought processes associated with toxic masculinity, specifically versus masculinity as a global construct. Using an instrument normed and validated that precisely measures toxic masculinity allows for more research that is more respectable and carries more weight in the literature, as a critical component of measurement in social science is the use of valid and reliable instruments [12]. Avoiding the use of proxy instruments when validated instruments are available contributes to both internal and external study validity in psychological research.

## 7. Limitations and Future Directions

There are several limitations to consider in this study. All participants were self-identified White men who were (at the time of data collection) current college students. This severely limits the ability to generalize the findings outside of this demographic. It would be beneficial to administer the measure in a community sample of men as well as intentionally recruiting a racially diverse sample of college-enrolled men. As masculinity is a construct heavily steeped in context, it is important to consider how groups differ on their conceptualization of both healthy and toxic masculinity. Future research would be well-served by addressing toxic masculinity in a thoughtful manner, fully considering cultural contexts. Additionally, as both men and women demonstrate masculine and feminine traits/behaviors, it may also be useful to administer the instrument to women to assess if the construct is uniquely expressed by men.

## Figures and Tables

**Table 1 behavsci-14-01096-t001:** 35 item Toxic Masculinity Scale Factor Structure, Items, internal reliability coefficients, and item factor loadings.

Item	Masculine Superiority	Domination and Desire	Gender Rigidity	Emotional Restriction	Repressed Suffering
Cronbach’s α	0.94	0.78	0.95	0.92	0.87
# of items	13	3	11	4	4
1. If I cry, I am weak				0.7904	
2. Crying means I am weak				0.8015	
3. Expressing sadness publicly makes me weak				0.7209	
4. Public displays of sadness means I am not masculine				0.6594	
5. Dominant men are attractive to their partners		0.7587			
6. People are attracted to men who dominate others		0.6938			
7. Muscles are indicators of masculinity		0.6917			
8. Men who wear women’s (e.g., dresses) clothing aren’t masculine			0.6551		
9. Men who wear dresses/skirts aren’t masculine			0.6512		
10. Men are superior to women	0.7197				
11. Gender and sex are the same thing			0.8599		
12. There are only two genders			0.8719		
13. Gender is not different from sex			0.8316		
14. It is propaganda that there are more than two genders			0.8718		
15. Men cheating on their partner is natural	0.6596				
16. Men should not work for women	0.7307				
17. My opinion as a man is more important than that of a woman’s	0.8296				
18. I don’t value the opinion of women	0.89				
19. A woman’s ideas are not as important as a man’s	0.8654				
20. Men should use only masculine pronouns (e.g., he/him)			0.7672		
21. Men can’t rape women because consent isn’t a real thing	0.6855				
22. Men’s behavior isn’t their fault because they are men	0.81				
23. Lying to my partner is okay, because I am a man	0.675				
24. If I am in pain, I don’t let people know					0.7087
25. I ignore pain when I feel it					0.8143
26. If I am sick, I refuse to go to the doctor					0.6899
27. If I don’t feel well, I just ignore it					0.7933
28. On average, men are smarter than women	0.7192				
29. Men are more capable than women	0.6688				
30. Men talking over women isn’t rude	0.7743				
31. I see it as a challenge when someone refuses my sexual advances	0.6795				
32. Sex workers have no self-respect			0.7125		
33. Sex work isn’t a career			0.7417		
34. There is no such thing as male privilege			0.6824		
35. Men don’t have inherent advantages in life			0.6569		

**Table 2 behavsci-14-01096-t002:** Comparison of goodness-of-fit indices for the 35- and 28-item versions of the novel TMS scale.

	*χ^2^*	*CFI*	*TLI*	*RMSEA*	*SRMR*	*AIC*	*BIC*
Administration 1 (35 items)	2948.59 (*df* = 550, *p* < 0.001)	0.77	0.76	0.11	0.09	30,745.62	31,061.47
Administration 1 (28-items)	1837.72 (*df* = 344, *p* < 0.001)	0.83	0.81	0.11	0.08	24,529.13	24,774.23
Administration 2 (35 items)	2333.22 (*df* = 550, *p* < 0.001)	0.79	0.77	0.11	0.10	21,674.22	21,962.10
Administration 2 (28 items)	1435.99 (*df* = 344, *p* < 0.001)	0.84	0.82	0.11	0.09	17,270.46	17,493.79

**Table 3 behavsci-14-01096-t003:** Phase 3 First Administration total scale correlations, means, standard deviations, and alpha coefficients (on the diagonal).

	1	2	3	4	M	SD
TMS	(0.94)				57.70	18.59
CMNI	0.73 ***	(0.90)			1.25	0.35
MNRI	0.81 ***	0.69 ***	(0.96)		2.75	1.21
LFAIS	−0.81 ***	−0.61 ***	−0.75 ***	(0.87)	3.14	0.89

Note. TMS = toxic masculinity scale; CMNI = conformity to masculine norms inventory; MNRI = male role norms inventory; LFAIS = liberal feminist attitudes and ideology scale; *** = *p* < 0.001.

**Table 4 behavsci-14-01096-t004:** Phase 3 Second Administration total scale correlations, means, standard deviations, and alpha coefficients (on the diagonal).

	1	2	3	4	M	SD
TMS	(0.95)				59.19	19.74
CMNI	0.75 ***	(0.92)			1.25	0.38
MNRI	0.830 ***	0.71 ***	(0.96)		2.89	1.23
LFAIS	−0.82 ***	−0.67 ***	−0.73 ***	(0.88)	3.09	0.93

Note. TMS = toxic masculinity scale; CMNI = conformity to masculine norms inventory; MNRI = male role norms inventory; LFAIS = liberal feminist attitudes and ideology scale; *** = *p* < 0.001.

**Table 5 behavsci-14-01096-t005:** Phase 3 1st administration TMS total scale correlations with 2nd administration instruments.

	1	2	3	4	5
TMS1	-				
CMNI2	0.68 ***	-			
MNRI2	0.77 ***	0.71 ***	-		
LFAIS2	−0.62 ***	−0.58 ***	−0.52 ***	-	
TMS2	0.93 ***	0.75 ***	0.83 ***	−0.64 ***	-

Note. TMS = toxic masculinity scale; CMNI = conformity to masculine norms inventory; MNRI = male role norms inventory; LFAIS = liberal feminist attitudes and ideology scale; *** = *p* < 0.001.

## Data Availability

Materials and analysis code for this study are available by emailing the corresponding author.

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
