# Peer review of "The Toxic Masculinity Scale: Development and Initial Validation"

_behavsci, 2024, doi:10.3390/bs14111096_

Round 1

Reviewer 1 Report

Comments and Suggestions for Authors

In my perspective, the work presented strictly follows all the necessary steps for the creation and initial validation of a scale. This scale, in turn, is highly pertinent to increase knowledge about psychosocial reality, through an increasingly used construct, but, as the authors point out, it lacks the most accurate (and measurable) delimitation in relation to similar constructs.

I highlight, as a very positive aspect, the author’s explanation of the limitations that must be recognized in this work. In fact, the circumscription of the study to white undergraduate men causes constraints on generalizability, which is recognized by the authors and, therefore, should be considered by researchers who will use this scale. However, it should be noted that the authors leave very relevant perspectives for the complexification and deepening of knowledge, referring to the need for future involvement of participants from other ethnicities, gender identities and cultural contexts.

The only recommendation I make refers to the concept of hegemonic masculinity - HM (page 2, 3rd paragraph). The importance of this concept for the development of studies on masculinity and its potential relationship with the concept (and some descriptive traits) of toxic masculinity deserve, in my perspective, a definition of HM and a reference to Connell and/or Kimmel.

Author Response

Thank you for reviewing our paper. We are excited about the future of this work, especially the validation process with male students of color and adult men who are not college students. We hope this work becomes generalizable and valuable for researchers broadly. 

To that end, we added multiple citations to strengthen our statements about hegemonic masculinity and added another line clarifying the definition. 

Reviewer 2 Report

Comments and Suggestions for Authors

The study addresses a very pressing issue in contemporary times - "toxic masculinity" - in an approach which emphasizes evidence-based data collected by means of surveys and their statistical evaluation and interpretation, wheres previous studies tackling this topic relied on auto-ethnographic statements and fieldwork results. My main reservation concerning this paper is precisely the main limitation mentioned in the Conclusion: the pool of informants being exclusively white (or "White" with a capital "W" as the authors refer to throughout the study) and college students. There are also some biographical elements which I would say need to be included in the bibliography, as I have noted in the attached file itself.

Author Response

Hello. Thank you so much for agreeing to review our manuscript. We appreciate your thoughtful feedback and additional information. It was clear you read our paper with intention. Please find our responses to your feedback below. 

  • We used APA formatting to guide the writing of this paper. According to the 7th edition of the APA style guide, the names of racial groups (e.g., White, Black, Asian) are capitalized. As a research team, we recognize that there are differing schools of thought on this topic, but as psychological researchers, we elected to follow the APA guidelines.
  • We added the period after Nicholas' middle initial.
  • We accepted the suggested word replacements in every instance
  • We moved the acronym definitions of the MRNI and CMNI to the second page and deleted the later instance of their explanations
  • The word womanist is a relatively recent addition to feminist theory emphasizing the experiences of Black women and their contributions to society. We believe this fits the theme of that section, but we are more than willing to revise it if you think it doesn't flow well
  • We added several citations in the section about hegemonic masculinity 
  • We added additional information about the Sona and Prolific platforms
  • We made significant changes to the implications section for clarity and conciseness